# The Detection of Immunity against WT1 and SMAD4^P130L^ of EpCAM^+^ Cancer Cells in Malignant Pleural Effusion

**DOI:** 10.3390/ijms232012177

**Published:** 2022-10-12

**Authors:** Terutsugu Koya, Yo Niida, Misa Togi, Kenichi Yoshida, Takuya Sakamoto, Hiroki Ura, Sumihito Togi, Tomohisa Kato, Sohsuke Yamada, Haruo Sugiyama, Shigeo Koido, Shigetaka Shimodaira

**Affiliations:** 1Department of Regenerative Medicine, Kanazawa Medical University, Kahoku 920-0293, Ishikawa, Japan; 2Center for Regenerative medicine, Kanazawa Medical University Hospital, Kahoku 920-0293, Ishikawa, Japan; 3Division of Genomic Medicine, Department of Advanced Medicine, Medical Research Institute, Kanazawa Medical University, Kahoku 920-0293, Ishikawa, Japan; 4Medical Research Institute, Kanazawa Medical University, Kahoku 920-0293, Ishikawa, Japan; 5Division of Stem Cell Medicine, Department of Advanced Medicine, Medical Research Institute, Kanazawa Medical University, Kahoku 920-0293, Ishikawa, Japan; 6Department of Pathology and Laboratory Medicine, Kanazawa Medical University, Kahoku 920-0293, Ishikawa, Japan; 7Department of Cancer Immunology, Graduate School of Medicine, Osaka University, Suita 565-0871, Osaka, Japan; 8Department of Gastroenterology and Hepatology, Jikei University School of Medicine, Kashiwa 277-8567, Chiba, Japan

**Keywords:** malignant pleura effusion, neoantigen, WT1, EpCAM, immunological memory

## Abstract

Malignant pleural effusion (MPE) provides a liquid tumor microenvironment model that includes cancer cells and immune cells. However, the characteristics of tumor antigen-specific CD8^+^ T cells have not been investigated in detail. Here, we analyzed MPE samples taken from a patient with pancreatic cancer who received a dendritic cell vaccine targeting Wilms’ Tumor 1 (WT1) antigen over the disease course (two points at MPE^1st^ and 2^nd^, two months after MPE1^st^). Epithelial cell adhesion molecule (EpCAM)^+^ cancer cells (PD-L1^−^ or T cell immunoglobulin mucin-3, TIM-3^−^), both PD-1 or TIM-3 positive CD8^+^ T cells, and CD14^+^CD68^+^CD163^+^TIM-3^+^ macrophages increased from the MPE^1st^ to MPE^2nd^. The ratio of WT1-specific cytotoxic lymphocytes (WT1-CTLs) to MPE CD8^+^ T cells and IFN-γ secretion of WT1-CTLs were reduced with disease progression. Coincidentally, the fraction of central memory T (T_CM_) of WT1-CTLs was decreased. On the other hand, CD8^+^ T cells in response to SMAD4^P130L^, which is homogeneously expressed in EpCAM^+^ cancer cells, were detected using in vitro expansion with the HLA-A*11:01 restrictive SVCVNLYH neoantigen. Furthermore, the CD8^+^ T cell response to SMAD4^P130L^ was diminished following remarkably decreased numbers of CD8^+^ T_CM_ in MPE samples. In conclusion, CD8^+^ T cells responding to WT1 or SMAD4^P130L^ neoantigen expressed in EpCAM^+^ pancreatic cancer cells were detected in MPE. A tumor antigen-specific immune response would provide novel insight into the MPE microenvironment.

## 1. Introduction

Immune checkpoint inhibitors (ICIs) that affect the immunosuppressive tumor microenvironment by recovering T cells have revolutionized cancer therapy. For example, ICIs block cytotoxic T-lymphocyte-associated antigen 4 (CTLA-4) and programmed death receptor-1/programmed cell death ligand-1 (PD-1/PD-L1) [1]. For the rapid establishment of effective ICI biomarkers, immune monitoring of cancer therapy efficacy should be performed using a malignant pleural effusion (MPE) model that comprises cancer cells and immune cells in a liquid microenvironment [2]. MPE could act as a biomarker for ICI therapy, chronologically detecting immune cell profiles during cancer progression, due to the dissemination of cancer cells with immune cells into a third space. Common sources of primary cancers causing MPE in the order of frequency include the lung, gastrointestinal tract, and pancreas in males, whereas the sources are the breast, lung, and ovary in females [3]. Although pancreatic cancer is the third common cause of MPE in males [3], pancreatic pleural effusion usually occurs in patients with acute or chronic pancreatitis and is rarely associated with pancreatic cancer [4]. As pancreatic cancer is suggested to be resistant to PD1 ICI with a low response rate of 3% [5], it is necessary to understand a pathological microenvironment using an MPE model to realize a revised immunotherapy approach for pancreatic cancer.

Immunological studies of the MPE environment have primarily focused on T cells expressing immune checkpoint molecules. The levels of checkpoint molecules such as PD-1/PD-L1 T cell immunoglobulin mucin-3 (TIM-3) and lymphocyte-activation gene 3 (LAG-3) were higher in CD4^+^ and CD8^+^ T cells of the MPE than in the peripheral blood [6]. It has been reported that MPE CD8^+^ T cells were revealed with a functionally decreased cytotoxic granule (granzyme B, perforin) and lower production of the effector molecule such as inflammatory cytokine and interferon-γ (IFN-γ) to kill neoplastic cells compared with CD8^+^ T cells in the peripheral blood [7]. Interestingly, CD4^+^ and CD8^+^ T cells in MPE derived from mesothelioma and lung cancer increase the population of central memory T (T_CM_) and effector memory T (T_EM_) cells as compared with those derived from peripheral blood [8]. However, the significance of this alteration of memory T cells for cancer immunity remains unaddressed. While immunosuppression of MPE has been elucidated by these characteristics of the T cell profile, monocyte/macrophages also exist in the MPE and contribute to the cancer microenvironment [9]. Therefore, hydrothorax-containing cancer cells with responding immune cells are a suitable model for cancer immune response analysis.

Memory T cells are essential for memorizing antitumor immunity to eradicate cancer [10]. CD8^+^ memory T cells have been elicited in response to tumor antigens such as Wilms’ Tumor 1 (WT1) and melanoma-associated antigen 3 (MAGE-A3) in the MPE [11]. In addition, neoantigens that occurred with the spontaneous mutation of the ROBO3 gene were identified in the cancerous hydrothorax of malignant mesothelioma [12]. This suggests that neoantigen-specific CD8^+^ memory T cells are likely included in MPE. Investigation of memory T cells specific to tumor antigens during disease progression might assist with identifying the immune microenvironment in MPE. This study aimed to reveal the CD8^+^ T cell response to WT1 and individual neoantigens from memory T cell subsets in the immune suppressive environment using an MPE model sampled from a patient with pancreatic cancer who received the dendritic cell vaccination targeting the WT1 antigen.

## 2. Results

### 2.1. Expression of Immune Checkpoint Molecules of EpCAM^+^ Cancer Cells, CD8^+^ T Cells, and CD14^+^CD68^+^ Macrophages in MPE Samples

We examined the expression of the immune checkpoint molecules on MPE cell types in a patient with advanced pancreatic cancer. Cancer cell markers, determined by epithelial cell adhesion molecule, and EpCAM^+^ cells [13] were observed in 0.60% and 1.28% of whole cells in the first (MPE^1st^) and second (MPE^2nd^) experiments (obtained two months after MPE^1st^), respectively (Figure 1A). The EpCAM^+^ cells were confirmed as adenocarcinoma using Papanicolaou staining. Immune checkpoint molecules such as PD-L1 and TIM-3 on EpCAM^+^ cancer cells could not be detected; on the other hand, a slightly increased PD-1- or TIM-3-positive CD8^+^ T cells was observed in MPE^2nd^ samples (PD-1^+^CD8^+^T cells: from 0.51% to 2.79%; TIM-3^+^CD8^+^T cells: from 3.26% to 5.24%, MPE^1st^ to MPE^2nd^, respectively) in Figure 1B. CD14^+^CD68^+^ macrophages in the MPE samples increased from 0.82% to 4.61% in CD45^+^ cells between the first and second sampling (Figure 1C). There was no PD-L1 expression on CD14^+^CD68^+^ macrophages; however, an increased CD163^+^TIM-3^+^ expression was observed in 58.1% and 69.1% of CD14^+^CD68^+^ macrophages in MPE1^st^ and MPE^2nd^, respectively.

### 2.2. WT1 Expression of EpCAM^+^ Cancer Cells and Characteristics of WT1-CTLs in MPE Samples

We identified the WT1 expression of the EpCAM^+^ cancer cells observed in MPE samples (Figure 2A). To closely mimic the MPE environment, both EpCAM positive and negative cells were sorted from the MPE^1st^ sample without cell culture. Immunostaining was performed using a WT1 monoclonal antibody. Despite the non-detection of EpCAM-negative cells, a green Alexa Fluor 488 fluorescence was detected in EpCAM^+^ cells, indicating a WT1 expression in cancer cells (Figure 2A, middle panel). WT1-specific cytotoxic lymphocytes (WT1-CTLs) defined by WT1-tetramer^+^CD8^+^ T cells were detected in MPE samples (Figure 2B, upper panel). The ratio of WT1-CTLs to CD8^+^ T cells in MPE decreased from 33.9% in MPE^1st^ to 16.8% in MPE^2nd^. Interestingly, the T_CM_ phenotype was indicated in most WT1-CTLs in MPE^1st^, whereas the T_CM_ phenotype was decreased in MPE^2nd^; and an increase in effector memory T (T_EM_) was found instead (T_CM_, from 82.6 to 53.7%; T_EM_, from 6.31 to 39.0% in WT1-CTLs, Figure 2B, middle panel). ELISpot assays to detect IFN-γ exposed to the WT1 antigen revealed a functional decrease in WT1-CTLs during disease progression (Figure 2B, lower panel). Thus, in a liquid immune suppressive microenvironment, both WT1-expressing EpCAM^+^ adenocarcinoma cells and WT1-CTLs reflected cancer progression.

### 2.3. Detection of SMAD4^P130L^ Expression on EpCAM^+^ Cancer Cells and CD8^+^ T Cells’ Response to HLA-A*11:01 Restricted Neoantigens for SMAD4^P130L^ in MPE Samples

We evaluated SMAD4^P130L^ expressing cells in MPE samples, which were identified from a long-range, PCR-based, dual deep sequence for targeting the Big 4 Genes KRAS, CDKN2A, TP53, and SMAD4 of pancreatic cancer [14,15] (Appendix A). A green Alexa Fluor 488 fluorescence was detected by immunostaining with a SMAD4 antibody, which indicated the expression of SMAD4 protein regardless of EpCAM expression on MPE cells (Appendix A). On the other hand, sequence analysis revealed a homozygous expression of *SMAD4* P130L in EpCAM^+^ cancer cells, but it failed for other EpCAM-negative fractions (Appendix A). Thus, protein expression of SMAD4^P130L^ was evident in EpCAM^+^ cancer cells. We then sought to determine if an immune response against a SMAD4^P130L^ antigen would be observed in the MPE samples. Based on the result of the in silico binding prediction of SMAD4^P130L^, and given the patient’s HLA class I, neoantigen candidate peptides were predicted with high affinity for HLA-A*11:01 (Table 1). Compared with a wild type, SMAD4-Neo1 and -Neo3 showed a high binding affinity (SMAD4-Neo1, 129 nM, SMAD4-WT1, 332 nM; SMAD4-Neo3, 193 nM, SMAD4-WT3, 416 nM in Table 1). To determine the CD8^+^ T response to these neoantigen candidates, an in vitro expansion of CD8^+^ T cells was performed, and sorted from the WT1 tetramer-negative fraction of the MPE^1st^, with antigen-presenting cells as autologous human platelet lysate–interferon–dendritic cells (HPL-IFN-DCs, HLA-A*11:01, and 24:02 types) containing each SMAD4 antigen. After the in vitro expansion, we confirmed the increases in IFN-γ-producing cells under the stimulation of SMAD4-Neo1 (8mer of SVCVNLYH) using ELISpot assays (Figure 3A). When SMAD4-Neo1 was converted to -WT1 (SVCVNPYH), there was no increase in IFN-γ-producing cells, suggesting a specific CD8^+^ T response to SMAD4-Neo1. We also sought to clarify the HLA-A*11:01 restriction that predicted the binding capability of SMAD4-Neo1. ELISpot assays were conducted using the HEV0271 cell strain as an antigen-presenting cell with a homozygote of HLA-A*11:01 harboring SMAD4-Neo1 peptides. CD8^+^ T cells purified from in vitro expanded cells with SMAD4-Neo1 showed increased IFN-γ-producing cells after stimulation with HEV0271 cell strain containing SMAD4-Neo1, but not with SMAD4-WT1 (Figure 3B). These results indicate that CD8^+^ T cells’ response to HLA-A*11:01-restricted SMAD4^P130L^ neoantigen existed in the MPE.

### 2.4. Detection of the Memory Subset of Infiltrated CD8^+^ T Cells in MPE That Responds to SMAD4^P130L^ during Disease Progression

From the CD8^+^ memory T cell subset in the MPE samples, the induction capability of SMAD4^P130L^ neoantigen-specific CD8^+^ T cells was evaluated. First, we identified the memory subsets of WT1 tetramer-negative CD8^+^ T cells in the MPE that potentially composed the SMAD4^P130L^ neoantigen-specific response. The WT1 tetramer-negative CD8^+^ T cells showed a remarkable decrease in T_CM_ with disease progression, i.e., from 43.4% MEP^1st^ to 7.49% MEP^2nd^ (Figure 4A). Furthermore, as a result of in vitro expansion using WT1 tetramer-negative CD8^+^ T cells with SMAD4-Neo1 antigen, CD8^+^ T cells’ response to SMAD4^P130L^ was decreased in MPE ^2nd^ compared to that in MPE ^1st^ (Figure 4B). Therefore, in MPE, CD8^+^ T_CM_ fluctuates relative to disease status, influencing the neoantigen-specific CD8^+^ T cells.
Figure 4Memory subsets of WT1 tetramer^−^ CD8^+^ T cells and detection of SMAD4^P130L^-specific CD8^+^ T cell expanded from MPE samples. (**A**) WT1 tetramer^−^ CD8^+^ T cells in MPE^1st^ or MPE^2nd^ were stained with antibodies against CD62L and CD45RO for memory T cell subsets. (**B**) WT1 tetramer^−^ CD8^+^ T cells were sorted from MPE^1st^ or MPE^2nd^, expanding in vitro with HPL-IFN-DCs containing SMAD4-Neo1 peptide. These cells were stimulated with SMAD4-Neo1 or DMSO to detect IFN-γ-producing cells by ELISpot assay. The ratio of IFN-γ spots of SMAD4-Neo1 to control was shown in the box plot (per each well in 96 plates). Mann–Whitney U test, * *p* < 0.05.
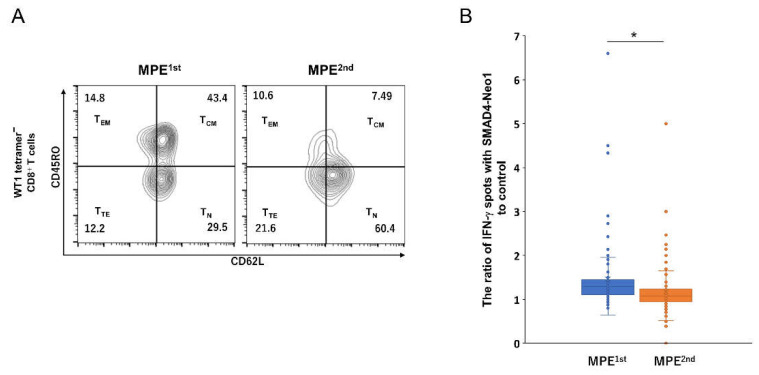



## 3. Discussion

EpCAM^+^ cancer cells (PD-L1^−^ or TIM-3^−^), CD8^+^ T cells with either PD-1^+^ or TIM-3^+^, and CD14^+^CD68^+^CD163^+^TIM-3^+^ macrophages increased in the MPE of a patient with progressive pancreatic cancer (Figure 1). Given that pancreatic cancer is associated with a low PD-L1 expression [16], a similar phenotype was observed in EpCAM^+^ cancer cells. Interestingly, CD14^+^CD68^+^ macrophages did not show PD-L1 expression, and an increased CD163^+^TIM-3^+^ expression was observed in MPE. In patients with hepatocellular carcinoma (HCC), TIM-3 expressions were increased in peripheral blood monocytes and tumor-associated macrophages (TAMs) at the tumor sites [17]. TIM-3 interference significantly inhibited the suppressive activities of macrophages in vitro and in vivo, promoting the suppression of HCC proliferation. This information would be helpful when investigating immunosuppressive activities through the interaction between TIM-3 on CD14^+^CD68^+^CD163^+^ macrophages in MPE and its ligands [18].

In this study, WT1 and SMAD4^P130L^ expressions in EpCAM^+^ cancer cells were found (Figure 2A, Appendix A), and the existence of CD8^+^ T cells, which respond to cancer antigens in MPE samples, was clarified (Figure 2B and Figure 3). High antitumor effects, with CD8^+^ T cells recognizing neoantigens, are especially expected [19]. However, an optimized identification of neoantigen-specific CD8^+^ T cells targeting a common neoantigen in heterogeneous cancer cells is vital if we induce effective antitumor reactions. A homogeneous expression of SMAD4^P130L^ might be a driver gene observed in EpCAM^+^ cancer cells in MPE (Appendix A). EpCAM^+^ cancer cells would be available for screening common neoantigens expressed in cancer cells with heterogeneous characteristics. EpCAM is upregulated in various primary tumors and metastases [20]. During epithelial-mesenchymal transformation, cancer cells might lose or lack their EpCAM expression [21]. Therefore, we must carefully validate whether the neoantigen screening from EpCAM^+^ cancer cells is commonly expressed in heterozygous cancer cells. Cancer genomic analyses might assist with the real-time, active monitoring of neoantigen-specific CD8^+^ T cells during disease progression using EpCAM^+^ cancer cells derived from MPE (Figure 1). A peripheral blood EpCAM^+^ circulating tumor cells (CTCs) is reported [22]. Alternatively, personalized neoantigens could also be detected in CTCs from patients with cancer for non-invasive treatment. Recent progress has suggested that circulating tumor DNA would be sensitive and specific liquid biopsy for the monitoring of patients with pancreatic cancer [23]. For screening neoantigens, these methods should be evaluated.

Immunity memory and booster effects can be confirmed with a WT1-DC vaccine [24,25]. The T_CM_ of WT-CTLs could evidently be detected in MPE (Figure 2B). We are currently conducting a clinical trial to evaluate the safety and feasibility of an HPL-IFN-DC vaccine harboring individualized neoantigens to confer memory immunity to neoantigen-specific T cells (Japan Registry of Clinical Trials, jRCTc040210109). This is a crucial issue for inducing cancer-associated antigen- or neoantigen-specific CD8^+^ T_CM_ to achieve a long-lasting antitumor immunity. The CD8^+^ T_CM_ in MPE remarkably decreased, with a reduction in CD8^+^ T cell response to WT1 and SMAD4^P130L^ (Figure 2 and Figure 4). At the same time, M2-like CD14^+^CD68^+^CD163^+^TIM-3^+^ increased (Figure 1C). M1 macrophages with TIM-3 upregulation at the tumor microenvironment polarize to an M2-type macrophage that can directly suppress antitumor immunity [26]. M1^hot^ TAMs boost tissue-resident memory (T_RM_) [27,28] infiltration and survival in patients with lung cancer [29]. Matos et al. demonstrated that circulating T_CM_ is a highly efficient precursor of human skin T_RM_ [30]. T_CM_ in MPE could also help generate T_RM_ for a long-lasting anticancer immunity. Further studies are necessary to determine if CD14^+^CD68^+^CD163^+^TIM-3^+^ contributes to anticancer immunity by controlling CD8^+^ memory T cells in MPE.

Pancreatic cancer is suggested to be resistant to PD1 ICI with a low response rate of 3% [5]. Because PD-L1-negative EpCAM^+^ cancer cells, low PD-1 expressed CD8^+^T cells, and remarkably increased CD14^+^CD68^+^CD163^+^TIM-3^+^ macrophages were observed in MPE with disease progression (Figure 1), TIM-3 blockade therapy might be applicable to treatment [31]. An MPE is expected to be an ideal model to reveal the mechanism underlying antitumor immunity in the immune suppressive microenvironment and the establishment of a biomarker for ICI therapy; however, there are several issues to be determined. To confirm our results universally, a large number of MPE cases with cancers should be evaluated. Moreover, the expression of neoantigens of EpCAM^+^ cancer cells in MPE should be monitored during disease progression, because a decreased CD8^+^ T cell response to SMAD4^P130L^ might occur due to the loss of the SMAD4^P130L^ antigen in MPE^2nd^. There was a loss of neoantigen expression on tumor cells because of immunoediting during cancer cell–T cell interactions in melanoma patients [32]. These findings would contribute to the progress of antitumor immunotherapy targeting further broad neoantigens.

## 4. Materials and Methods

### 4.1. Ethics Statement and Cellular Materials from the Pleural Effusion

The dendritic cells (DC) vaccination study targeting WT1 was conducted at Shinshu University Hospital and approved by the Ethics Committee of Shinshu University School of Medicine (approval number 1199, 2 December 2008; 2704, 8 April 2014). A patient with pancreatic cancer met the eligibility criteria for DC vaccination therapy and was determined to be HLA-A*24:02 and A*11:01 DNA types. He underwent three courses of WT1-pulsed DC vaccination and developed progressive pleural involvement with effusion 5 months after the final DC vaccine was administered in July 2018. We obtained two samples of the patient’s MPE, taken for the primary purpose of reducing respiratory discomfort, between July 2018 and September 2018. At the time of collecting the second MPE sample, we noticed that the patient’s cancer had progressed remarkably. All cellular materials were collected after obtaining written informed consent following the Declaration of Helsinki approved by the Ethics Committee of Kanazawa Medical University (approval number G156; 8 June 2020).

### 4.2. Phenotyping of MPE Cells

MPE samples were centrifuged at 500× *g* at 4 °C for 5 min to collect cells. Each aliquot of MPE cells was incubated with the mouse IgG anti-human monoclonal antibodies, as shown in Appendix A. After incubation for 1 h at 4 °C, cells were washed with FACSFlowTM (BD Biosciences, Franklin Lakes, NJ, USA) and were examined for phenotyping. For intracellular staining of CD68, we used IntraPrep (Beckman Coulter, Inc., Miami, FL, USA) for fixation and permeabilization according to the manufacturer’s instructions.

### 4.3. Immunofluorescence Staining

EpCAM positive and negative live cells were sorted from MPE samples using a Cell Sorter SH800 (Sony Biotechnology Inc., Tokyo, Japan). Immediately after sorting, a cell slide was prepared using a Smear Gell Kit (GenoStaff Co., Ltd., Tokyo, Japan) according to the manufacturer’s instructions. Subsequently, 1 × 10^5^ cells were fixed with a 10% formalin neutral buffer solution (Wako Pure Chemicals Ltd., Osaka, Japan) for 30 min and then permeabilized with a 0.1% Triton X-100 (Sigma-Aldrich Co. LLC, St. Louis, MO, USA) solution in PBS at room temperature for 5 min and blocked with UltraCruz Blocking Reagent (Santa Cruz Biotechnology, Inc., Dallas, TX, USA) for 60 min. After blocking, the cells were incubated with a mouse monoclonal primary anti-WT1 antibody (1:100, clone 6F-H2, Thermo Fisher Scientific, Waltham, MA, USA) at 4 °C for 16 h. Subsequently, a goat anti-mouse IgG (H+L) highly cross-adsorbed secondary antibody conjugated with Alexa Fluor plus 488 (1:200, Thermo Fisher Scientific) was added at room temperature for 30 min. Finally, the cells were incubated with DAPI (Thermo Fisher Scientific) at 300 nM in PBS at room temperature for 2 min.

### 4.4. Memory T Cell Subsets and Functional Analysis of WT1-CTLs

MPE cells were stained at 4 °C for 30 min with antibodies in Appendix A. Enzyme-linked immunospot (ELISpot) assays were performed using precoated human IFN-γ ELISpot PLUS kits (Mabtech AB, Nacka Strand, Sweden) according to the manufacturer’s protocol. In total, 1 × 10^4^ MPE cells were seeded in 96-well plates in the presence of 10 μM WT1-235 peptides (CYTWNQMNL, residues 235–243; PEPTIDE INSTITUTE, INC., Osaka, Japan) in AIM-V medium (Thermo Fisher Scientific) supplemented with 10% FBS (Thermo Fisher Scientific).

### 4.5. Neoantigen Prediction for SMAD4^P130L^

Neoantigen prediction for SMAD4^P130L^ based on the binding affinities of 8– to 11–mer peptides to HLA class I molecules (HLA-A, -B, and -C) were examined using NetMHC v3.4 software and NetMHCpanv2.8, as described previously [33]. The top three peptides with IC50 < 500 nM were synthesized by GenScript USA Inc. (Piscataway, NJ, USA) and subjected to further analysis (Table 1).

### 4.6. CD8^+^ T Response to SMAD4^P130L^ Neoantigen

The CD8^+^ T cells sorted from WT tetramer-negative fraction in the MPE samples using a Cell Sorter SH800 (Sony) were applied for cultivation. Autologous generated human platelet lysate (HPL) interferon-α-induced DCs (HPL-IFN-DCs) [34] containing each SMAD4**^P130L^** neoantigen peptide compatible for HLA-A*11:01 types were used as antigen-presenting cells. These cells were co-cultured in a 96-well U-bottom plate at a ratio of 1:10 (HPL-IFN-DCs:CD8^+^ T cells) and in vitro expansion using previously reported protocol [34]. After in vitro expansion, 5 × 10^5^ cells were stimulated with SMAD4^P130L^ predictive neoantigen peptides for IFN-γ targeting ELISpot assays. Alternatively, the human B lymphocyte transformed by the Epstein–Barr virus, and the HEV0271 cell strain (HLA-A*11:01 homozygous, provided by the RIKEN BRC through the National BioResource Project of the MEXT/AMED, Japan), were applied as an antigen-presenting cell pulsed with SMAD4-Neo1 or -WT1 peptide at 37 °C for 2 h. The CD8^+^ T cells purified with CD8 microbeads (Miltenyi Biotec) after in vitro expansion under HPL-IFN-DCs containing SMAD4–Neo1 were used as responder cells. For 16 h, 1 × 10^4^ CD8^+^ T cells were co-cultured with HEV0271 at a 1:1 ratio in AIM-V supplemented with 10% FBS to detect IFN-γ spots.

### 4.7. Statistical Analysis

Unpaired *t*-tests were performed to compare independent group means. In addition, Dunnett’s test was applied to compare multiple groups to a single control group. The nonparametric analyses were the Wilcoxon Signed Ranks Test for paired data and the Mann–Whitney *U* test for unpaired data. All statistical analyses were performed using IBM SPSS Advanced Statistics software, version 23.0 (IBM Japan, Tokyo, Japan). Differences were considered statistically significant at a *p*-value < 0.05.

## 5. Conclusions

CD8^+^ T cells responding to WT1 or SMAD4^P130L^ neoantigen expressed in very few EpCAM^+^ pancreatic cancer cells were detected in MPE. We also observed increases in CD14^+^CD68^+^CD163^+^TIM-3^+^ in MPE that corresponded to disease progression. Therefore, the blockade of TIM-3 on CD14^+^CD68^+^CD163^+^ macrophages might improve cancer immunosuppression. An MPE model detecting a tumor antigen-specific immune response under an immunosuppressive microenvironment would be useful for the development of antitumor immunotherapy.

## Figures and Tables

**Figure 1 ijms-23-12177-f001:**
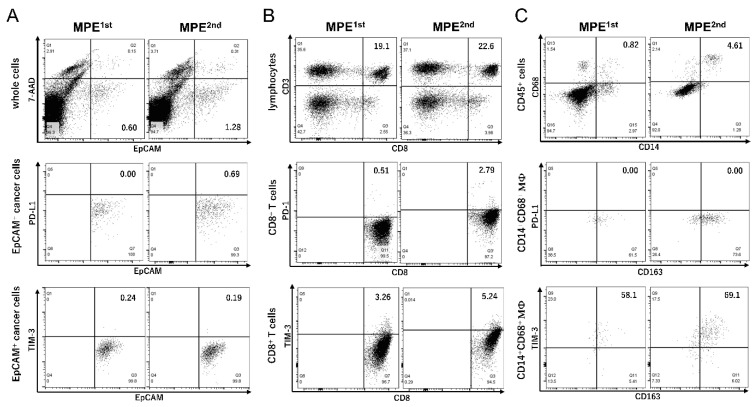
In MPE samples, there are immune checkpoint molecules on EpCAM^+^ cancer cells, CD8^+^ T cells, and CD14^+^CD68^+^ macrophages. (**A**) Representative flow cytometry plots for PD-L1 or TIM-3 expressions of EpCAM^+^ cancer cells in malignant pleura effusion 1st and 2nd (MPE^1st^, MPE^2nd^). (**B**) CD8^+^ T cells in CD45-SSC gating lymphocytes (MPE^1st^, 19.1%; MPE^2nd^, 22.6%) and PD-1 or TIM-3 expressions on CD8^+^ T cells in MPE. (**C**) Percentage of PD-L1 or TIM-3 expressions on CD14^+^CD68^+^ macrophages (MΦ) in CD45-SSC gating cells in MPE.

**Figure 2 ijms-23-12177-f002:**
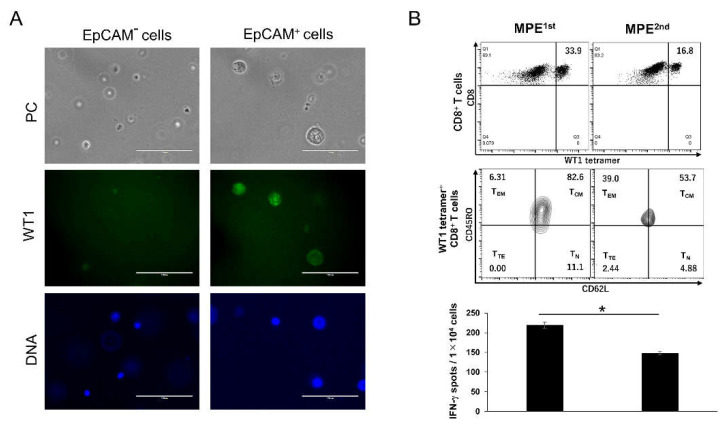
WT1 expression on EpCAM^+^ cancer cells and detection of memory subsets of WT1-CTLs in MPE samples. (**A**) Expression of WT1 on EpCAM positive or negative cells sorted from whole cells in MPE^1st^. PC, phase contrast; WT1 cells were stained with anti-WT1 antibody, followed by Alexa Fluor® Plus 488 secondary antibody; DNA, DAPI staining. The white bar indicates 100 μm. (**B**) The ratio of WT-CTLs is defined as WT1 tetramer^+^ CD8^+^ T cells to MPE samples^1st^ or MPE^2nd^ (upper panel). The memory subsets of WT1-CTLs during disease progression (middle panel). CD62L^+^CD45RO^+^: central memory T cells (T_CM_); CD62L^+^CD45RO^−^: naïve T cells (T_N_); CD62L^−^CD45RO^+^: effector memory T cells (T_EM_), CD62L^−^CD45RO^−^: terminal effector T cells (T_TE_). The function of WT1-CTLs in MPE was evaluated by ELISpot assay for IFN-γ secretion under WT1 235 peptide stimulation (lower panel). The mean number of spots of WT1 235 peptide-specific was shown. The error bar indicated the mean and standard deviation. Unpaired *t*-test, * *p* < 0.05.

**Figure 3 ijms-23-12177-f003:**
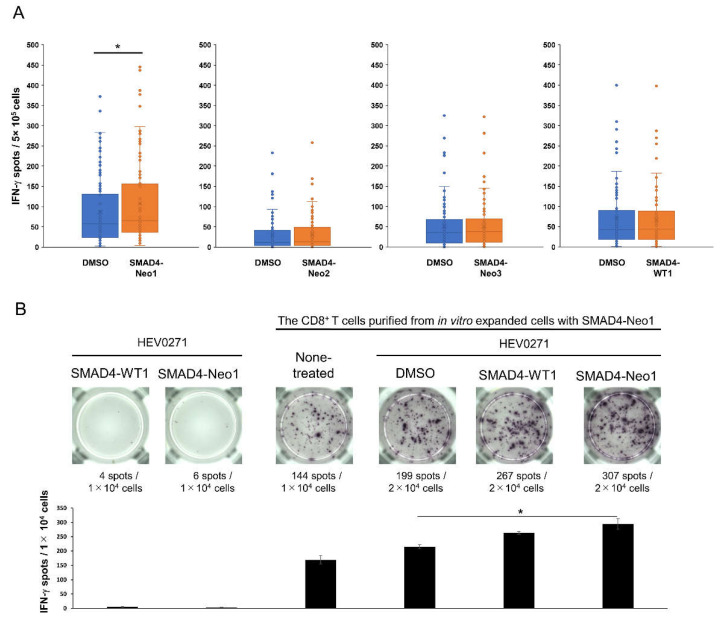
Detection of CD8^+^ T cell response to HLA-A*11:01 restricted SMAD4^P130L^ neoantigen in MPE. (**A**) WT1 tetramer-CD8^+^ T cells sorted from MPE^1st^ were expanded with HPL-IFN-DCs containing each predicted neoantigen peptide for SMAD4^P130L^. For ELISpot assay to detect IFN-γ production after in vitro expansion, these cells were stimulated using each SMAD4^P130L^ peptide. Reactivity was shown in the box plot (per each well in 96 plates). DMSO was used as a negative control. A Wilcoxon signed-rank test was performed, * *p* < 0.05. (**B**) CD8^+^ T cells purified from in vitro expanded cells with HPL-IFN-DCs contained SMAD4-Neo1 peptides in Figure 4A were stimulated by HEV0271 cells (HLA-A*11:01 homozygous) containing SMAD4-Neo1 (SVCVNLYH) or SMAD4-WT1 (SVCVNPYH). The reactivity to IFN-γ was evaluated by ELISpot assay. The error bar indicated the mean and standard deviation. Dunnett’s test, * *p* < 0.05.

**Table 1 ijms-23-12177-t001:** Top five predicted SMAD4^P130L^ neoantigens to HLA class I.

Mutant Peptide		Wild-Type Peptide			
Sequence	Affinity to HLA (nM)	Sequence	Affinity to HLA (nM)	Amino Acid Length	HLA-Type
SVCVNLYH	129	SVCVNPYH	332	8	HLA-A*11:01
CVNLYHYER	180	CVNPYHYER	84	9	HLA-A*11:01
SVCVNLYHY	193	SVCVNPYHY	416	9	HLA-A*11:01
SVCVNLYHYER	285	SVCVNPYHYER	419	11	HLA-A*11:01
SVCVNLYHY	224	SVCVNPYHY	634	9	HLA-B*15:01

The top three mutant peptides were named as follows: SMAD4-Neo1: SVCVNLYH, SMAD4-Neo-2: CVNLYHYER, and SMAD4-Neo-3: SVCVNLYHY. Wild-type peptides corresponding to these individual neoantigen candidates were designated as follows: SMAD4-WT1: SVCVNPYH, SMAD4-WT2: CVNPYHYER, and SMAD4-WT3: SVCVNPYHY.

## Data Availability

The data presented in this study are available in the article or Appendix A.

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
