# Peer review of "The Detection of Immunity against WT1 and SMAD4P130L of EpCAM+ Cancer Cells in Malignant Pleural Effusion"

_ijms, 2022, doi:10.3390/ijms232012177_

Round 1

Reviewer 1 Report

The authors examined the expression of WT-1 and SMAD4 (P130L) in EpCAM-positive cancer cells in malignant pleural effusions (a case of pancreatic cancer), as well as the status of CD8+ T cells that respond to these.

Samples were pleural effusions collected at 2-month intervals from one case.

EpCAM-positive tumor cells are WT-1 positive, EpCAM-negative are negative. In CD8+ CTL, T central memory was abundant in the first sample, and switched to T effect memory in the second sample.

Furthermore, the authors also examined the position of SMAD4 as a tumor antigen, and showed that P130L was expressed in EpCAM+ cancer cells, and that CTLs expanded with this antigen reacted only with HLA11:01.

Based on these results, the authors suggest that the amount or response of CTLs in pleural effusion can serve as a tumor marker for dynamic immune pathology in the progression of disease (pancreatic cancer).

The first is an analysis from the pleural effusion of one case of pancreatic cancer, which is very suggestive data, but it is unclear whether the same thing will occur in other cases.

Furthermore, first of all, considering the frequency of malignant pleural effusion in pancreatic cancer, is it really a marker? "What is the advantage compared to other markers?" If pleural puncture is also required, it will be very invasive.

It is strongly desired that at least the results of investigations of key points in several cases of similar tumors be taken into account.

Author Response

We thank you greatly for your reply and review our manuscript (Manuscript ID: ijms-1913863). We appreciate the comments provided by the reviewer that have allowed us for further improvement of our manuscript. We have carefully revised the manuscript following the reviewer's suggestion. We also extensively revised our manuscript for better understanding including typographical correctness. All changes have been made in a red character.

Reviewer 1

The authors examined the expression of WT-1 and SMAD4 (P130L) in EpCAM-positive cancer cells in malignant pleural effusions (a case of pancreatic cancer), as well as the status of CD8+ T cells that respond to these.

Samples were pleural effusions collected at 2-month intervals from one case.

EpCAM-positive tumor cells are WT-1 positive, EpCAM-negative are negative. In CD8+ CTL, T central memory was abundant in the first sample, and switched to T effect memory in the second sample.

Furthermore, the authors also examined the position of SMAD4 as a tumor antigen, and showed that P130L was expressed in EpCAM+ cancer cells, and that CTLs expanded with this antigen reacted only with HLA11:01.

Based on these results, the authors suggest that the amount or response of CTLs in pleural effusion can serve as a tumor marker for dynamic immune pathology in the progression of disease (pancreatic cancer).

The first is an analysis from the pleural effusion of one case of pancreatic cancer, which is very suggestive data, but it is unclear whether the same thing will occur in other cases.

  1. As reviewer’s kind suggestion, we clearly revised this sentence in lines 252-255 of the revised manuscript.

Furthermore, first of all, considering the frequency of malignant pleural effusion in pancreatic cancer, is it really a marker? "What is the advantage compared to other markers?" If pleural puncture is also required, it will be very invasive.

  1. As reviewer’s kind suggestion, we clearly revised this sentence in lines 2-3, 44, 82, 84, and 341-343 of the revised manuscript. We removed the statement of biomarker and concluded that an MPE model detecting tumor antigen-specific immune response under immune suppressive microenvironment would be useful for the development of antitumor immunotherapy.

It is strongly desired that at least the results of investigations of key points in several cases of similar tumors be taken into account.

  1. As reviewer’s kind suggestion, we clearly revised this sentence in lines 56-59, 252-261 of the revised manuscript.

Reviewer 2 Report

Koya et al., have presented an interesting case study using MFE from the pancreatic cancer patient treated with DC vaccine against SMAD and WT1 neoantigen. All the experiments are well designed and executed. The statistics is rigorous. Althogh,  the data is generated  from a single patient, is important to study response and biology of a DC vaccine treated patients. This observations and study design can be potentially used for other studies that are using different vaccines for different cancers. Overall, the study is very good. However, authors should consider the following to reflect on the title and discussion of the manuscript.

1.      There is no control MFE (from patient who didn’t receive DC vaccine) analyzed over time hence concluding that  “tumor antigen- specific immune response based on memory CD8+ T cell presumably acts as a dynamic immunological biomarker during cancer progression” is an over statement. It is not clear whether it is due to normal course of cancer progression or it is an immunological biomarker indicative of progression. Same holds true for CD14+CD68+CD163+TIM-3 + in MPE that corresponded to disease progression. Since there is no control MFE that is coming from the patient that did not receive DC vaccine it is hard to make this conclusion. Either authors should do this analysis with the control patients or tone down the conclusion and the title.

2.      The authors need to comment about antigen escape since reduction in T cells in the second MFE and expansion of suppressive macrophages coincides with tumor progression. Can this be due to antigen escape?

Author Response

We thank you greatly for your reply and review our manuscript (Manuscript ID: ijms-1913863). We appreciate the comments provided by the reviewer that have allowed us for further improvement of our manuscript. We have carefully revised the manuscript following the reviewer's suggestion. We extensively revised our manuscript for better understanding including typographical correctness. All changes have been made in a red character.

Reviewer 2

Koya et al., have presented an interesting case study using MFE from the pancreatic cancer patient treated with DC vaccine against SMAD and WT1 neoantigen. All the experiments are well designed and executed. The statistics is rigorous. Although, the data is generated from a single patient, is important to study response and biology of a DC vaccine treated patients. This observations and study design can be potentially used for other studies that are using different vaccines for different cancers. Overall, the study is very good. However, authors should consider the following to reflect on the title and discussion of the manuscript.

1.There is no control MFE (from patient who didn’t receive DC vaccine) analyzed over time hence concluding that “tumor antigen- specific immune response based on memory CD8+ T cell presumably acts as a dynamic immunological biomarker during cancer progression” is an over statement. It is not clear whether it is due to normal course of cancer progression or it is an immunological biomarker indicative of progression. Same holds true for CD14+CD68+CD163+TIM-3 + in MPE that corresponded to disease progression. Since there is no control MFE that is coming from the patient that did not receive DC vaccine it is hard to make this conclusion. Either authors should do this analysis with the control patients or tone down the conclusion and the title.

As reviewer’s kind suggestion, we clearly revised this sentence in lines 2-3, 44, 82, 84, and 341-343 of the revised manuscript. We removed statement of biomarker in revised manuscript.

2. The authors need to comment about antigen escape since reduction in T cells in the second MFE and expansion of suppressive macrophages coincides with tumor progression. Can this be due to antigen escape?

As reviewer’s kind suggestion, we clearly revised this sentence in lines 242-243 and 255-261 of the revised manuscript.

Round 2

Reviewer 1 Report

A Revised manuscript has been posted.

Among them, the word "ratio" is used, for example, as "ratio of WT1-CTLs in CD8+ T cells", but ratio is originally used as follows.

According to Oxford Advanced Learner's Dictionary

”the relationship between two groups of people or things that is represented by two numbers showing how much larger one group is than the other

-The school has a very high teacher-student ratio.

 -ratio of A to B: What is the ratio of men to women in the department?

-The ratio of applications to available places currently stands at 100:1.”

Such re-checking of English is required.

The authors refer to Malignant pleural effusion as a model, which is only a rare, extensive form of pancreatic cancer. Although this case can be evaluated as an attempt, it seems that there were tumor cells in the peripheral blood in this case as well.

After comparing the CD8+ T cells in the pleural effusion and the CD8+ T cells in the peripheral blood, is it possible to try using circulating tumor cells in the peripheral blood?

When using malignant pleural effusion as a model, is it possible to apply this approach to cases such as lung cancer? (Is this type of analysis meaningless in cancers where immune checkpoint inhibitors are effective to some extent?)

After all, how to universalize the results of this time? In addition, although pancreatic cancer may be targeted, I would like you to mention how to make it a method that can be used even in rare cases such as those with malignant pleural effusion.

Author Response

We thank you deeply again for your reply and review our manuscript (Manuscript ID: ijms-1913863). We appreciate the comments provided by the reviewer that have allowed us for further improvement of our manuscript. We have carefully revised the manuscript following the reviewer's suggestion. We added the sentence of EpCAM+ CTCs and ctDNA for screening of neoantigen, interpretation of our results relevance to clinical development. To confirm our results universally, it should be evaluated a large number of cases MPE with cancers. All changes have been made in a red character.

A Revised manuscript has been posted.

Among them, the word "ratio" is used, for example, as "ratio of WT1-CTLs in CD8+ T cells", but ratio is originally used as follows. 

According to Oxford Advanced Learner's Dictionary

”the relationship between two groups of people or things that is represented by two numbers showing how much larger one group is than the other

-The school has a very high teacher-student ratio.

 -ratio of A to B: What is the ratio of men to women in the department?

-The ratio of applications to available places currently stands at 100:1.” 

Such re-checking of English is required.

As reviewer’s kind suggestion, we correct the sentence in lines 35, 100-101, 123, 138 of the revised manuscript.

The authors refer to Malignant pleural effusion as a model, which is only a rare, extensive form of pancreatic cancer. Although this case can be evaluated as an attempt, it seems that there were tumor cells in the peripheral blood in this case as well. 

After comparing the CD8+ T cells in the pleural effusion and the CD8+ T cells in the peripheral blood, is it possible to try using circulating tumor cells in the peripheral blood?

As reviewer’s kind suggestion, we clearly revised this sentence in lines 240-245 of the revised manuscript.

When using malignant pleural effusion as a model, is it possible to apply this approach to cases such as lung cancer? (Is this type of analysis meaningless in cancers where immune checkpoint inhibitors are effective to some extent?)

As reviewer’s kind suggestion, we clearly revised this sentence in lines 56-61, 262-269 of the revised manuscript.

After all, how to universalize the results of this time? In addition, although pancreatic cancer may be targeted, I would like you to mention how to make it a method that can be used even in rare cases such as those with malignant pleural effusion.

As reviewer’s kind suggestion, we clearly revised this sentence in lines 262-269 of the revised manuscript.

Round 3

Reviewer 1 Report

Authors modified their manuscript according to the reviewers' comments.